# An Electro-Thermal Actuation Method for Resonance Vibration of a Miniaturized Optical-Fiber Scanner for Future Scanning Fiber Endoscope Design

**Aydin Aghajanzadeh Ahrabi [1], Mandeep Kaur [1], Yasong Li [1], Pierre Lane [2] and Carlo Menon [1,\*]**

[1] Menrva Research Group, Schools of Mechatronic System Engineering and Engineering Science, Simon Fraser University, M250-13450 102 Avenue, Surrey, BC V3T 0A3, Canada; aaa91@sfu.ca (A.A.A.); mka116@sfu.ca (M.K.); yla83@sfu.ca (Y.L.)

[2] BC Cancer Research Center, Imaging Unit, Integrative Oncology, Room 6-122 (office), 5-208 (lab), 675 West 10th Avenue, Vancouver, BC V5Z 1L3, Canada; plane@bccrc.ca

\* Correspondence: cmenon@sfu.ca

**Abstract:** Medical professionals increasingly rely on endoscopes to carry out many minimally invasive procedures on patients to safely examine, diagnose, and treat a large variety of conditions. However, their insertion tube diameter dictates which passages of the body they can be inserted into and, consequently, what organs they can access. For inaccessible areas and organs, patients often undergo invasive and risky procedures—diagnostic confirmation of peripheral lung nodules via transthoracic needle biopsy is one example from oncology. Hence, this work sets out to present an optical-fiber scanner for a scanning fiber endoscope design that has an insertion tube diameter of about 0.5 mm, small enough to be inserted into the smallest airways of the lung. To attain this goal, a novel approach based on resonance thermal excitation of a single-mode 0.01-mm-diameter fiber-optic cantilever oscillating at 2–4 kHz is proposed. The small size of the electro-thermal actuator enables miniaturization of the insertion tube. Lateral free-end deflection of the cantilever is used as a benchmark for evaluating performance. Experimental results show that the cantilever can achieve over 0.2 mm of displacement at its free end. The experimental results also support finite element simulation models which can be used for future design iterations of the endoscope.

**Keywords:** electro-thermal actuation; resonance vibration; micro-cantilever beam; optical fiber scanner; simulation; design experiments

## 1. Introduction

Visualization of internal organs using endoscopy is the most effective and direct method to localize lesions that require further diagnostic tests or treatment. Endoscopes come in various configurations, each with different features and each optimized for the portion of the body they are designed to examine; however, the basic shape and the layout of the instrument are relatively unchanged since they were first introduced. Almost all include a long flexible insertion tube, an optical-fiber light delivery system to illuminate the organ being inspected, a lens system to collect the reflected light, and an image sensor—either a charge-coupled device (CCD) or complementary metal–oxide–semiconductor (CMOS) array detector [1]. Although endoscopic technologies provide real-time video of the organ surface with relatively high resolution, they require direct access to the internal body cavity or organ being examined. Thus, what medical procedures can be performed by endoscopes on which parts of body or organs are dictated by the insertion tube diameter. Furthermore, the insertion tube diameter of the endoscope should be as small as possible, to minimize discomfort to the patients and reduce

incision size and recovery time [2]. The rigidity of the endoscope should also be as low as possible for ease of maneuvering inside a patient's body [2].

Endoscopes based on CCD and CMOS sensors offer excellent performance and produce high-quality and high-resolution videos. However, the smaller the endoscope is, the smaller the solid-state sensor is and, consequently, the smaller the pixel elements on the sensor have to be, which results in lower sensitivity to light and higher noise levels [3,4]. Therefore, as the light-sensing elements get smaller, their sensitivity and their resolution (the number of resolvable pixels elements) are reduced. This limits how small the endoscope can get, and there is a significant drop in image quality of endoscopes that are less than 3 mm in diameter [5]. New advancements were made in recent years in the form of scanning fiber endoscopes (SFE) to further miniaturize the insertion tube diameter and this resulted in relatively high-fidelity endoscopes that are only about 1.2 mm [5–10]. The main component of the SFE is a single laterally vibrated resonant (first peak) single-mode optical-fiber scanning laser light onto the image plane. The higher the amplitude of vibrations is, the wider the field of view is (FOV). The time-multiplexed backscatter signal can then be collected by one (dual-clad) [11–14] or more (single mode) [5,7–9,15] optical fibers acting as detectors. Actuation methods of the illuminating fiber vary and can include lead zirconate titanate (PZT) piezoelectric actuators [5–15], aluminum nitride (AlN) piezoelectric thin films [16,17], and silicon on insulator (SOI) thermal electronics [18,19]. However, these actuation methods again become a limiting factor in how small the SFEs can get. Piezoelectric actuators produce very little amounts of deflection, requiring any piezoelectric actuator to use a stack of them to produce any meaningful displacement (~0.4 mm in most cases). In addition, these devices rely on numerous external multimode collector fibers situated in or around the housing, which make up a significant portion of the 1.2-mm insertion tube diameter. Nevertheless, this does not allow access to the smallest peripheral airways of the lung where the majority of suspicious nodules are most likely to occur.

Herein, a thermally actuated dual-clad fiber is proposed to address the two main factors preventing SFEs from shrinking further in size: their actuation method, and their collection fibers. This work introduces a novel actuation method in the form of thermal actuation to achieve the desired lateral vibratory motion. In addition, a dual-clad optical-fiber usage is proposed for both illumination and collection of reflected light, permitting to create an endoscope system small enough to be packaged inside of a sub-500-μm distal housing—small enough to access the smallest distal airways of the lung. The detailed design and fabrication process is presented in Section 2. Finite element analysis, given in Section 3, is performed to predict and explain the device's behavior and performance. Section 4 gives the experimental results, which show that the design is indeed small enough to meet the size requirements, while having a sufficient level of performance. Recommendations for future work are laid out in Section 4 as well. Section 5 provides a summary of the project and draws conclusions by highlighting the main findings.

## 2. Actuator Design and Fabrication Process

Cantilever design follows the finite element analysis done to investigate the dynamic behavior of a micro-cantilever that is transversely excited at its base using a free-standing metallic actuator that is expanding and contracting due to thermal cycles, reported in our previous works [20–24]. Figure 1 shows a three-dimensional (3D) model of the thermal actuator, dual-clad optical fiber, and fiber cantilever, mounted in the bore of a hypodermic needle. As shown in the figure, the fiber is placed along the long side of the metal actuator, which relates to the light collection from the inner clad of the fiber in the future design. However, from a mechanical and assembly point of view, the perpendicular alignment is more desired because it is easier to assemble, and it allows the actuator to be closer to the base, resulting in more tip displacement. Parallel alignment is chosen as the default set-up as it will obstruct less of the collection fiber than the perpendicular alignment.

The sub-500-μm outer diameter size limitation is the defining factor and the single most important characteristic of this proposed endoscope enclosure, and many design decisions arise from it. At the

core of this design lies a single-mode dual-core optical fiber custom-made by the authors. In order to fabricate this fiber, a multi-mode optical fiber (core diameter significantly larger than that of the dispersion compensating fiber) is heated up and pulled (by the same technique used to taper fibers for fiber couplers) to form a neck. When the neck narrows to the desired diameter (elaborated on in Section 2.2), it is cut and spliced to the core of a dual-core fiber ($\phi$ = 125 µm). The thin portion of the fiber, referred to as the micro-cantilever ($\phi$ = ~11 µm) is the only part being excited and vibrating. The thin cantilever section (~2 mm length) is resting on top of a metal bridge constructed out of single piece of laser-cut thin foil. The fiber itself is excited from the base by the fine wire shape (metal bridge) that is expanding and contracting due to a periodic current passing through it. The periodic signal through the fine wire matches the natural frequency of the vibrating fiber to induce resonance (first peak). The fine wire needs to be more resistive than the conducting wires so that it heats up more than them; however, at the same time, it has to be very thermally conductive to cool down in time during the off portion of the cycle at such high frequencies. The fine wire is cooled down by the larger conducting portions which act as heat sinks. Thermal conduction is the dominant means of heat transfer at this scale and frequency; effects of convection and radiation are negligible [23]. Optimal bridge dimensions are given in a previous work [23], i.e., 20–25-µm-diameter cross-sectional area and at least 100-µm bridge height to provide the desired level of performance of over ~200 µm of tip displacement, measured at the end of the cantilever at a frequency of at least 2 kHz. The pieces are fixed in place using polymer collars made by soft lithography before being inserted into the ~500-µm-diameter tube housing. This fiber scanner can scan along a line on one axis. The hypodermic needle can rotate at certain speed relying on a fiber-optic rotation joint. With both the vibration and rotation movement of the fiber core, the illumination laser can cover a circular area in a radial pattern (see inset of Figure 1). A cap which also holds the lens system at its tip is used to seal the tube.

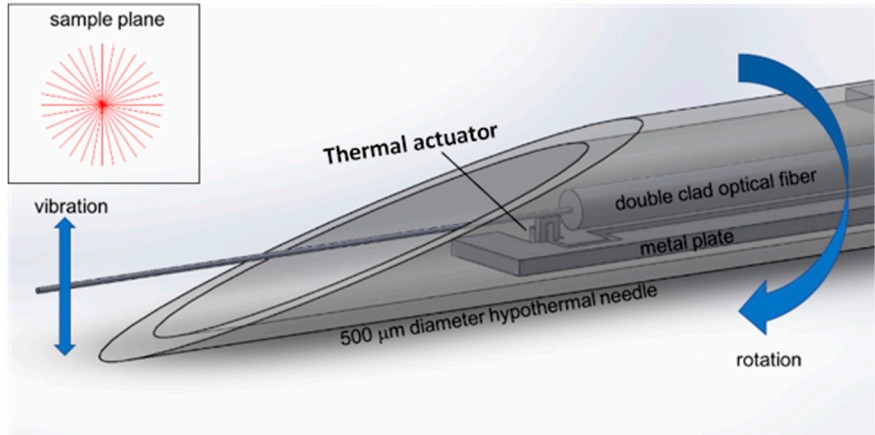

**Figure 1.** Thermal actuator, dual-clad optical fiber, and micro-cantilever optical fiber mounted in the bore of a hypodermic needle.

### 2.1. Material Selection

For a comprehensive and systematic material selection process, software package "Granta: CES Selector" was used to find suitable materials for the electro-thermal actuator. The main criteria considered for the actuation wire were the coefficient of linear thermal expansion, thermal conductivity, Young's modulus, and electrical resistivity. The coefficient of linear thermal expansion is an important factor that dictates how much heat needs to be generated by resistive heating in the fine wire to achieve the desired level of actuation. Thermal conductivity determines how fast the heat is conducted through fine actuating wire. If this number is not sufficiently high, the actuating wire will not cool down to the starting temperature before the start of the next cycle, thus reducing the temperature difference between the on and off states and, consequently, affecting the lateral vibrations of the fiber in a negative manner. Young's modulus is used to determine how well the actuation wire can withstand

the vibrations of the fiber without bending, buckling, or deforming. In this project, a high electrical resistivity in the actuation wire is desired because it means a lower current value is required to heat up the wire to the desired temperatures. For conduction wires, lower resistivity and high thermal conductivity are the desired selection parameters.

After a review of the range of parameters amongst various metals, it was observed that the coefficient of linear expansion does not vary significantly. Instead, electrical resistivity and thermal conductivity are the parameters that can wildly fluctuate. Therefore, the materials that did not meet the minimum specified Young's modulus were filtered out (at least 70 GPa). Then, the selection process prioritized thermal conductivity first and foremost, followed by electrical resistivity and, finally, coefficient of linear expansion. Note that, generally, metals that have excellent thermal conductivity generally have low electrical resistivity values. This poses a problem because, if the actuation and conduction wires have the same resistance, for the same given current, the actuation wire would have a marginally higher temperature and, therefore, a lower temperature differential. For this reason, the group of metals that had moderate electrical resistance (over 2.5 μΩ·cm) and acceptable levels of thermal conduction (over 120 W/m·K) were chosen. Table 1 shows the properties of various candidate materials to be used in the prototypes.

**Table 1.** Comparison of the selected metals.

|  | Aluminum 1199 (Pure) | Aluminum 6061 | Brass 7030 | High-Tensile Manganese Brass |
|---|---|---|---|---|
| Linear thermal expansion (μstrain/°C) | 22.4 | 24.6 | 19 | 21 |
| Thermal conductivity (W/m·K) | 220 | 170 | 126 | 121 |
| Young's modulus (GPa) | 75 | 71 | 110 | 91 |
| Electrical resistivity (μΩ·cm) | 2.5 | 4.4 | 6.5 | 23 |

This selection process identified aluminum–silicon–copper alloys (aluminum 6 series alloys) followed by brass alloys and pure aluminum (aluminum 1 series) as suitable materials for the actuation wire. Aluminum 6061 has a slightly lower thermal conductivity, but it is not so low that it would start to affect its performance at our frequencies. However, it is significantly more resistive than pure aluminum and it has a higher thermal expansion coefficient. Even though yellow brass (brass 7030) is marginally worse than aluminum 6061, in terms of its material properties, it is a high alloy of copper which means it can easily be joined to the conducting copper wires via a multitude of methods, giving it a significant edge in ease of use. All three materials, aluminum 6061, aluminum 1199, and brass 7030, are used as the bridge material in various samples. For the conduction wires which also double as heat sinks, pure copper is used.

## 2.2. Resonance Frequency

The principle behind the actuation method in this design is the resonance phenomenon, i.e., the tendency of a mechanical structure to vibrate at a great amplitude when it is excited at its natural frequency. The first mode shape, the fundamental frequency, is targeted to yield the desired large displacement of the tip of the optical fiber. The natural frequency also determines the sample rate of the system and, therefore, should be as high as possible. Mechanical resonance of an object is a function of its shape and size. To achieve the desired frequency for the system, the dimensions of the vibrating optical micro-cantilever have to be changed. As expected, the system behaves very similarly to a fixed-free cylindrical cantilever and can be modeled as such. The general form of the equation to solve for the natural frequency is

$$1 + \cos(\beta L) \cosh(\beta L) = 0, \tag{1}$$

with

$$\beta^4 = \frac{m\omega^2 + \frac{\pi}{4}\rho\omega^2 b^2 \Gamma(\omega)}{E_{beam}I}, \qquad (2)$$

where $L$ is the length, $m$ is the mass, $E$ is the modulus of elasticity, $\rho$ is the density, $b$ is the dominant scale length (width in the case of a rectangular section, and diameter for a circular section), $I$ is the second moment of inertia, and $\Gamma$ is a dimensionless function [22].

This allows us to map the natural frequency of the fiber as a function of its length and diameter, as shown in Figure 2. The shorter the fiber is, the higher the frequency will be. Similarly, a larger micro-cantilever diameter results in a higher natural frequency. Ideally, the actuator is placed as close to the base of the cantilever as possible. Although a high degree of oscillations is desired, it comes at a cost. A higher natural-frequency value decreases the time between each thermal cycle, and, at some point, the system is not going to have enough time to cool back down to the starting temperature, which can lead to a rising average temperature and lower the magnitude of tip displacement. Furthermore, a larger diameter and a shorter length increase the flexural rigidity of the micro-cantilever and decrease the amplitude of the lateral vibratory motion. Therefore, the dimensions of the optical fibers are chosen in a way that the resonance frequency is maximized while the desired level of tip displacement is maintained. As a result, the samples made and tested have a resonance frequency of about 2.3–3.4 kHz and are about 1.7–2.1 mm in length and about 11 μm in diameter.

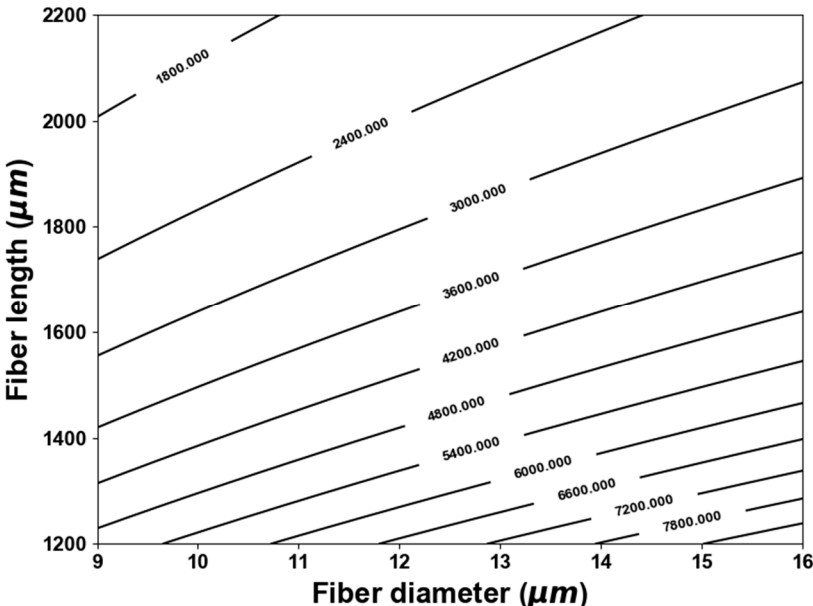

**Figure 2.** Damped natural frequency of the system (in Hz) as a function of micro-cantilever length and diameter.

### 2.3. Thermal Actuator

In order to fabricate the free-standing actuator bridge, in a way that is repeatable and reliable, in the desired dimensions, the actuator bridge is laser-cut out of metal thin foils using a laser micromachining workstation (IX-280-ML, IPG photonics, Oxford, MA, USA). The bridge is then manually lifted 90° to an orthogonal position to create the upright bridge. The bridge is kept upright with the help of plastic deformation of the cut-out portion. Electrical resistivity increases proportionally when the cross-sectional area decreases. Therefore, the resistive actuating wire can be made by cutting a narrow strip out of a larger thin metal foil, and the metal foil that is still connected to it has a larger cross-sectional area and a relatively lower electrical resistivity. This means that the larger portion does not heat up as much as the narrow strip while maintaining a perfect electrical and thermal interface with it. This thin foil is then connected to the conducting copper wires which act as heat sinks and help

transfer the generated heat away from the actuator during the off cycles via conduction (see schematic in Figure 3a). Although copper and aluminum are dissimilar metals, notoriously difficult to join together, and are susceptible to problems such as galvanic corrosion and brittle resulting joints [25], they can be connected together using methods such as single-point tab bonding, capacitive-discharge spot welding, and brazing or soldering using aggressive flux. Brass, being a high alloy of copper, is significantly easier to work with and can be easily joined to the copper lines via spot welding or soldering. Therefore, the thin foils being laser-cut are made of both aluminum (44233 aluminum foil, Alfa Aesar, Ward Hill, MA, USA) and brass (13505 brass foil, Alfa Aesar) foils and are 25 μm thick. In the computer-aided design (CAD) files, the actuating wire is supposed to be 25 μm wide; however, because of the laser cutter cutting into the line by a few microns, the resulting strip is about 22 μm thick, which causes the actuating wires to have a cross-sectional area of $22 \times 25$ μm$^2$. This resulting rectangular bridge shape addresses many alignment, fixing, and positioning issues and allows the optical fiber to be carefully positioned on top of the metal bridge, ideally as close to the base as possible (see Figure 3b). This method is very versatile in terms of creating the desired bridge shape and controlling its dimensions, and it is used to create rectangular bridges that are 60-, 100-, 150-, and 200-μm-tall samples.

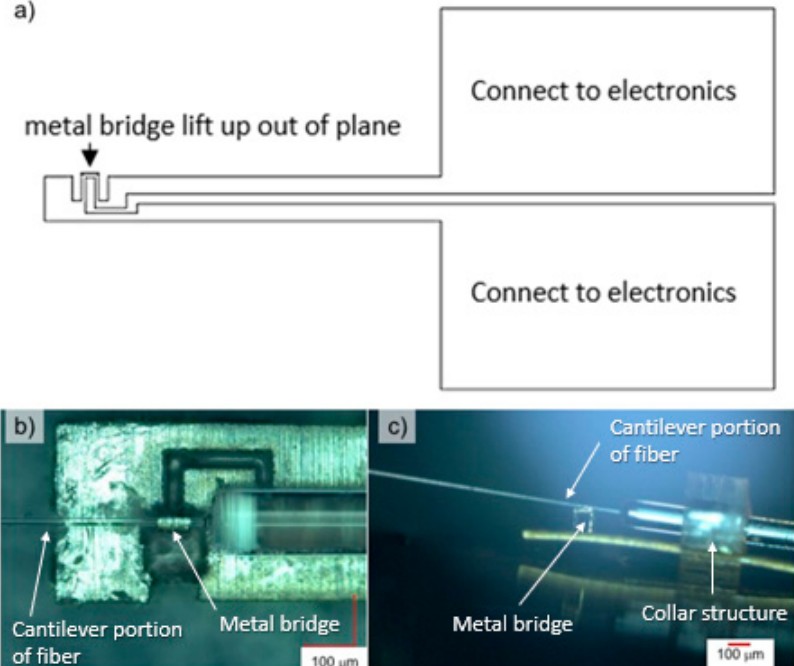

**Figure 3.** Assembly process of the main components of thermal actuator: (**a**) schematic showing the pattern cut out of the metal thin foil to create the metal actuator bridge; (**b**) top view of the optical fiber resting on the metal actuator; (**c**) side view of the assembled system, with a collar structure to fix components in place.

*2.4. Housing, Packaging, and Adhesives*

The main goal of this project is to make an endoscope system that is small enough to fit inside of a sub-500-μm housing. Therefore, it was determined that it is difficult to assemble the device inside of the tube, and everything has to be assembled and fixed in place first and then inserted into the tube. To achieve this goal, a series of polymer "collar" structures were fabricated using photolithography (SU-8 3050, MicroChem, Westborough, MA, USA), to package the device. The semi-circle piece is meant to create a flat surface/platform for the metal thin foil cut-out, while the outer diameter of the top collar is made to be smaller than the inner diameter (~400 μm) of the tube (see plastic parts in Figures 3c and 4a,b), so that it can be inserted into the tube. The pieces are then glued together using a

heat-cured epoxy (EP17HTND-CCM, Master Bond, Hackensack, NJ, USA). Once inside, more epoxy is injected into the tube and cured to fill out any gaps between the components and also stick the inserted components to the inside of the tube. Multiple collars are often used next to each other to create a wider support zone. The hypodermic needle is laser-cut at a sharp angle to make it possible to observe and monitor the device during operation and testing.

The adhesive used for this application has to be able to bond metal, glass, and various polymers together. Furthermore, they should not shrink and expand during and after the curing process because that would misalign the fiber and bridge; furthermore, it should be hard enough that it does not dampen the vibrations (an absolute necessity for a fixed-free cantilever model), and it must have a high glass transition temperature so that it does not soften at elevated temperatures when the device is running. These criteria severely limit the choice of adhesive. For this selection process, all basic classes of adhesives were studied or tested, and only some epoxy and acrylic adhesives remained. After further examination, the heat-cured epoxy by Master Bond Inc. was chosen due to its standout properties such as ~85 Shore D hardness and a glass transition temperature of 235 °C.

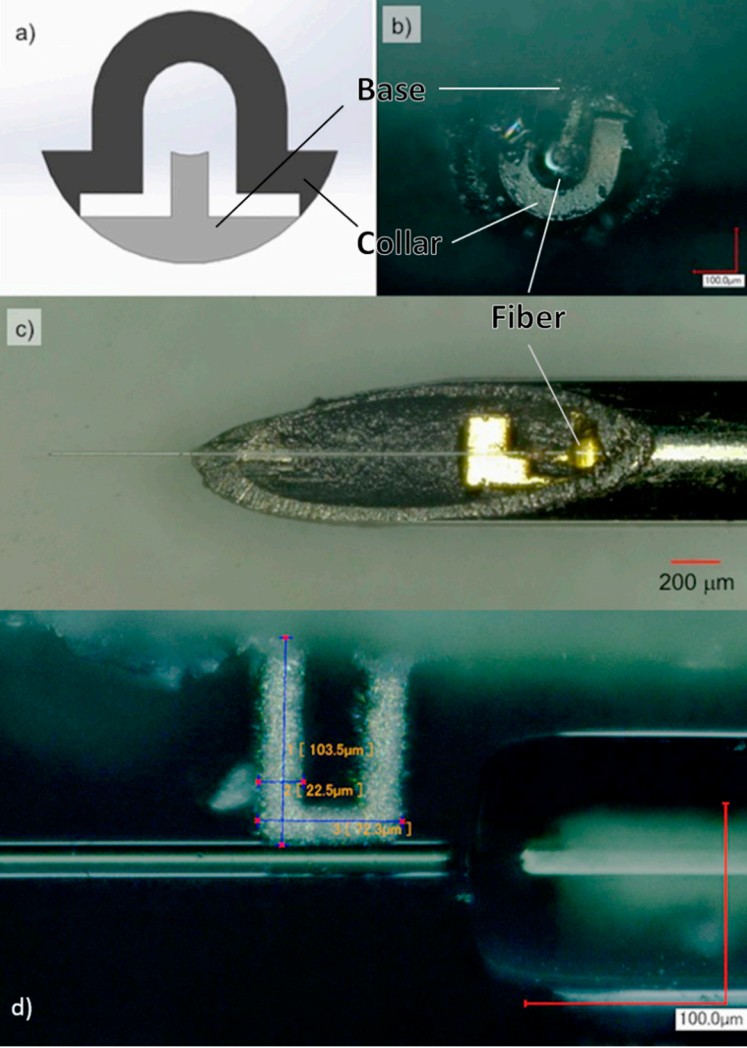

**Figure 4.** Collar structure to assist positioning of the scanner assembly inside the hypodermic needle: (**a**) schematic showing the two-part collar structure; (**b**) front view of the assembly (upside down relative to (**a**)), clearly showing the collar structure placed in the middle of the hypodermic needle; (**c**) top view of the assembly—the collar is hidden in this view as it is inside of the housing tube; (**d**) zoomed-in view of a 100-μm-tall aluminum actuator bridge with geometric parameters.

The design decisions taken in this section show that it is possible to have a package that has an outer diameter of less than 500 μm. In addition, this combines for a rigid distal tip length of only about 3 to 5 mm. These results validate the proposed fabrication procedures in practice (See Figure 4).

## 3. Simulation for Optimization

Finite element analysis of the exact model (see Figure 5) used in the actuator system was performed to predict the response of the system and make suggestions to optimize its performance, specifically fiber-tip displacement and temperature at the vibrating bridge. The simulations were performed in four parts using ANSYS (Version 18.2, ANSYS Inc., Canonsburg, PA, USA), and COMSOL Multiphysics (Version 5.4, COMSOL AB, Stockholm, Sweden).

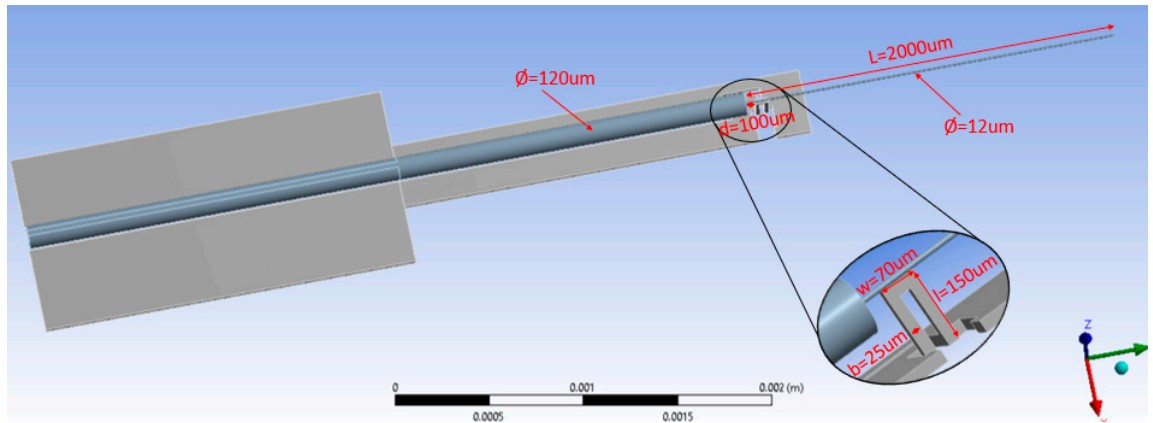

**Figure 5.** Dimensions of the device being simulated using ANSYS.

### 3.1. Fiber-Tip Displacement vs. Time

Fiber-tip displacement was simulated at resonance frequency of the optical fiber using COMSOL Multiphysics. Modal analysis determined the resonance frequency of the optical fiber, which was set as an input for the simulation. To simulate the damping effect, air was set to surround the device (see Figure 6). Aluminum 6061 was used as the actuator (bridge) material and was set to be 100 μm tall. In a specific instance of the simulation, a signal in the form of a square wave cycle at 1370.916 Hz with a current value of 1.04 A ran through the structure. The length of the cantilever fiber was about 1.85 mm and the diameter was 11 μm; the distance between the vibrating bridge and the junction of large and small fiber was 100 μm. The fiber-tip displacement in the above set-up was estimated to peak and settle at ~80 μm unlike the undamped response.

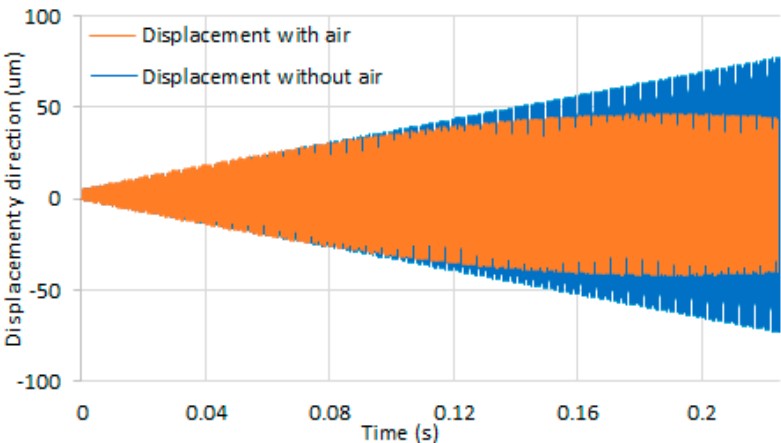

**Figure 6.** Resonance time response of the cantilever fiber measured at the tip, with the damping effect of surrounding air versus no damping.

### 3.2. Bridge Temperature vs. Time

As an electro-thermal actuator, at steady state, the higher and lower bounds of the temperature of the actuating bridge dictate how much it will expand and contract, which in turn dictates how well it oscillates the optical fiber placed on top of it. Furthermore, at elevated temperatures, prolonged use of the device can heat up the surrounding environment which can quickly soften and degrade the quality of adhesive, which in turn impairs the mechanical integrity of the entire device (less rigid) and lowers performance because of the increased damping on the "fixed" end of the system. Thus, it is imperative to know how hot the actuator gets and how its temperature fluctuates. The temperature variations of the actuator due to an electric pulse were studied with respect to time. Figure 7 shows the simulation of temperature of an aluminum 6061 actuating bridge with an average input power of 0.314 W and a square signal at 2 kHz, at room temperature of 20 °C, and the body temperature of 37 °C. The temperature quickly settled at ~53 °C and ~70 °C, respectively, in 0.12 s. Temperature fluctuated within ~3 °C (see inset of Figure 7) because of the cyclic current passing through the actuating bridge which was essentially a resistive load. The temperature fluctuations were contained within ~3 °C in the simulated period. The time for the temperature to reach steady state was roughly the same as the time for the settlement of the tip displacement (see Figure 6). The system response was generally the same for a higher electric current input, albeit with a higher steady-state temperature.

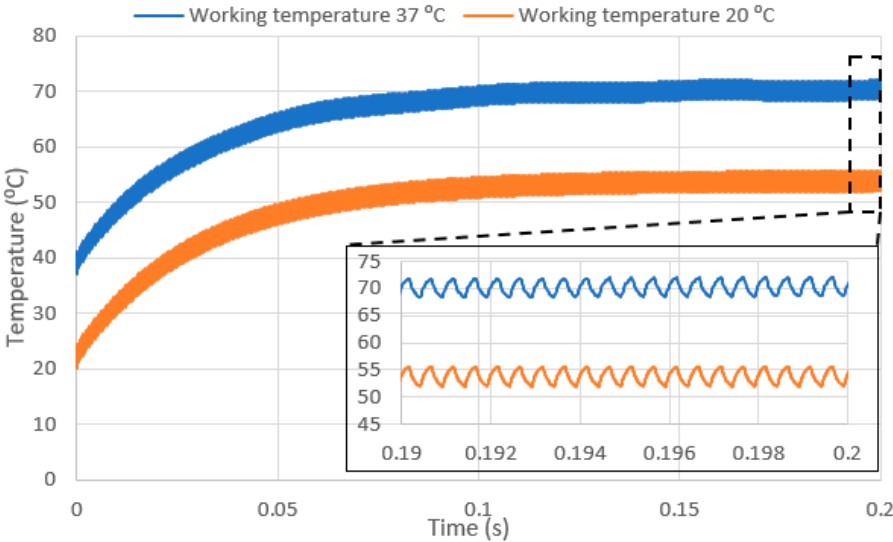

**Figure 7.** Temperature profile of the aluminum actuating bridge at 0.314-W power input. The inset enlarges the temperature profiles from 0.19 s to 0.2 s.

### 3.3. Bridge Temperature vs. Input Current

With a higher input current, more energy flows through the actuating bridge and causes a higher bridge temperature and larger thermal expansion. The bridge temperature under various power inputs was simulated to determine if the temperature rises to a high enough degree that it might compromise the structural integrity of the system and damage any of the components. Figure 8 shows the simulated aluminum bridge temperature at different current levels using ANSYS. The frequency of electric pulses given to the bridge was 3 kHz. Literature shows that the Young's modulus decreased from ~4 GPa to ~2.5 GPa when SU-8 was heated up to 50 °C from room temperature (25 °C); the Young's modulus further decreased to ~1 GPa when SU-8 was heated to 100 °C [26]. The temperature increased to 80 °C when it was given a high current input (1.4 A), which could possibly cause the collar section to soften. The collar section is intended to keep the fiber and metal foil in place, so that the fiber can achieve desired oscillation performance. Soft collars may cause undesirable damping effects which can decrease the amount of lateral tip displacement of the oscillating fiber. For all previous simulations, the electric pulse given to the metal bridge was a square wave with 50% duty cycle. Decreasing the

duty cycle, i.e., giving the bridge more time to cool down during the off cycles, becomes a solution to lower the risk of mechanical damping effect due to temperature rise.

It was also observed from the simulation that the higher the current level was, the larger the temperature fluctuation at steady state was (0.12 s–0.2 s, see Figure 7). The temperature fluctuation increased from ~0.5 °C to ~4 °C, for a current level of 0.47 A to 1.41 A (see Figure 8); the temperature differential was higher at higher power levels.

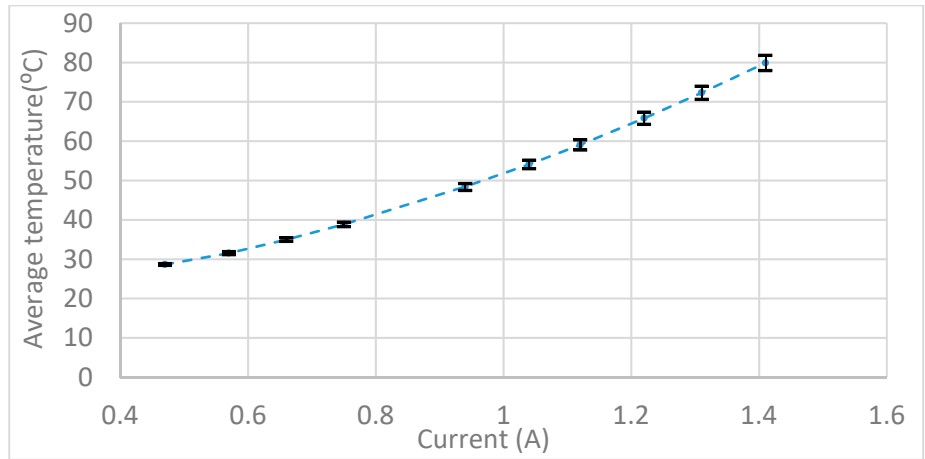

**Figure 8.** Temperature response of the aluminum bridge at different current levels using an actuation frequency of 3 kHz.

## 4. Experimental Validation and Discussion

The goal of testing the prototypes was to evaluate their response to periodic input from the actuator bridge in the form of tip displacement and to characterize these oscillations as a function of power. Given the fact that actuator bridges can come in different dimensions and thicknesses and can be made of various metals and, therefore, can have different resistance values, power, measured in watts, is the unit of measurement for input. Furthermore, since the oscillations are caused by the expansion and contraction of the bridge due to cyclic resistive heating, using power as a unit for input is very suitable. This also makes it possible to have something that can be compared when comparing two different samples together. Prototypes can be simply modeled as time-invariant resistive loads.

A metal-oxide semiconductor field-effect transistor (MOSFET)-based driver circuit was used to pass a square wave through the actuator bridge. With the help of pull-up resistors and adjusting the input voltage, the power through the bridge can be controlled. The prototypes have a resistance value of ~0.25 Ohms. A Tektronix AFG1062 function generator was used to generate cyclic input waves. A digital optical 1000× magnification microscope (Keyence VHX-2000, Keyence, Osaka, Japan) was used to observe and monitor the tip displacement of prototypes; this was enough to provide high-resolution images that could be used to accurately measure the movement of the 11-μm-diameter fiber to sub-micron levels. Before starting the test procedure, the natural frequency of the sample was found by slowly sweeping the frequency range where the fiber was expected to have its natural frequency (based on its dimensions) while the tip of the fiber was observed. The exact natural frequency of the system was found when the tip of fiber showed an increase in the amplitude of vibrations. After the resonance frequency of the prototype was found, the input power was set to a lower voltage and then ramped up in small increments. The response of the system at each step was recorded to form a complete picture of the harmonic response of the prototype.

### 4.1. Harmonic Response

The main premise of this study is the resonance vibration of a micro-cantilever under cyclic thermal excitation. In a structural system, any cyclic load will produce a sustained cyclic or harmonic

response. Therefore, it is important to verify this claim and prove that the oscillations are indeed amplified significantly at the first natural frequency by doing a harmonic analysis. The results are used to determine the steady-state response of a linear structure to loads that vary harmonically with time, normally done to avoid and overcome resonance; however, in this case, resonance is the desired outcome. For this, the input power was kept the same, but the input frequency was swept slowly from a point before the natural frequency and then past it. The frequency sweep was linear. The harmonic response of the system, i.e., the induced lateral vibrations of the tip, were recorded at small increments. The harmonic response of a laser-cut 100-μm aluminum sample with a natural frequency of 2381 Hz and a duty cycle of 50% is shown in Figure 9. A noticeably greater amplitude of displacement was observed when the frequency of input oscillations matched the micro-cantilever's natural frequency than it did at other frequencies. The harmonic response of different prototype configurations was about the same with the only notable variations being the natural frequency (as the prototypes were handmade) and amplitude of displacements; the slope and the quality factor stayed consistent. The quality factor remained low across the board at approximately 4. This indicates a high damping coefficient. Factors that can contribute to damping in the system are the hardness of the adhesive holding the "fixed" end of the fiber, the actuator bridge flexing and absorbing some of the vibrations (only ~20 μm thick and relatively soft as a result), how off axis the fiber is relative to the actuator bridge, any softening that might occur in the SU-8 collars as a result of rising average temperatures, etc. Therefore, any improvements in the system that might increase the structural rigidity of the device are highly desirable. Increased rigidity of the system will increase the quality factor, which in turn will increase the lateral tip displacement proportionally.

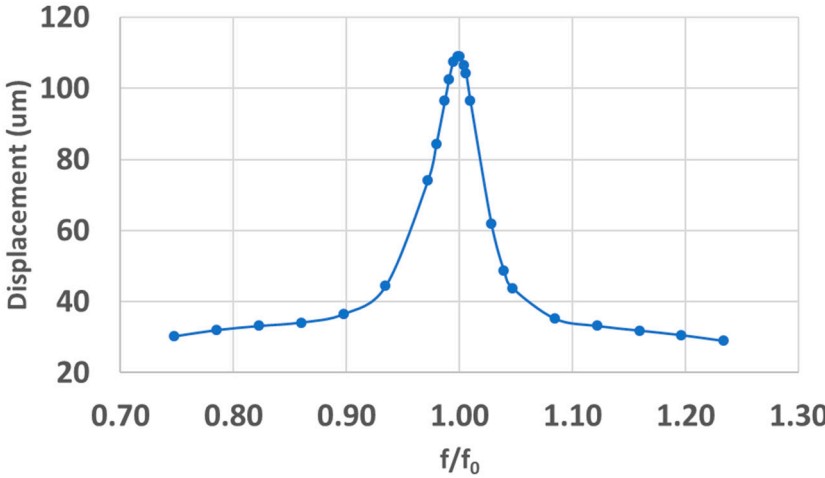

**Figure 9.** Typical harmonic response of a 100-μm-tall aluminum bridge sample with a natural frequency of 2381 Hz (50% duty cycle).

### 4.2. Tip Displacement at Natural Frequency

To assess the response of the system, in the form of magnitude of tip displacement, to increased input power, the input frequency was kept the same as the system's natural frequency and power through the actuator was gradually ramped up. The magnitude of tip displacement at every interval was recorded. The input power was kept well below 1 W. The tip displacement response of a 100-μm-tall aluminum bridge with a natural frequency of 2381 Hz and a 150-μm-tall aluminum bridge with a natural frequency of 2410 Hz are shown in Figure 10. The distance from the base of the cantilever fiber to the actuator was about 100 μm for both samples. Since coefficient of thermal expansion is linear, a taller actuator should result in a higher level of displacement, proportional to its increase in height. In theory, a 50% increase in tip displacement should be observed by increasing the height of the bridge from 100 μm to 150 μm. However, this increase in bridge height resulted in an approximate increase of about 23% in displacement. This can be due to many factors that would cause this deviation from

the ideal case. A 150-µm bridge is approximately 40% more resistive than a 100-µm bridge because of its increased length and mass; therefore, a larger body of mass from a higher temperature has to be cooled down in the same time cycle. This increased resistance also causes a lateral shift in input power for the same current input. A 150-µm-tall bridge would also be less structurally rigid than a shorter one, and can flex and bend more, reducing the performance by having a higher damping coefficient and lower quality factor.

Many of the prototypes fabricated used brass as the bridge material because aluminum is notoriously hard to work with, both due to being dissimilar to copper and being extremely soft in a 20-µm-thin foil form factor. All four of the prototypes in Figure 11 had a 150-µm-tall brass bridge. Two samples had a distance of 100 µm from the base of the micro-cantilever, while the other two had a distance of 50 µm and no gap. As expected, the brass samples with their higher resistivity and lower thermal conductivity and linear expansion performed worse than their aluminum counterparts; the 150-µm-tall brass samples were on par with the 100-µm-tall aluminum samples. The increase in resonance frequency corresponded to only about a 100-µm reduction in length. Even though the shorter micro-cantilever should have a smaller amplitude of lateral vibration than the longer fiber, the effect should not be as pronounced. It was evident that a sample with a higher frequency had a significant decrease in performance. This was observed in many samples. This can be attributed to the shorter time cycle in which the actuator bridge has to cool down. This increased resonance frequency can cause a higher average temperature at steady state, which would mean a lower cyclic temperature differential and less vertical oscillations by the actuator bridge. These results mainly point to a heat dissipation issue. This is further reinforced by looking at the data from 200-µm-tall brass-bridge samples. These samples only provided a marginal improvement in performance compared to their 150-µm counterparts, and, after a certain point in input power, they not only started to plateau, but they actually regressed, proving they are not cooling fast enough. As seen in Figure 11, the brass sample with the 50-µm distance and a natural frequency of 3008 Hz had a fairly linear response until about the 0.350-W mark, where it started to plateau and regress. This is clear evidence of the bridge not cooling down sufficiently fast enough and, thus, having a higher initial temperature before the next cycle and, consequently, a lower temperature differential. In addition, having the actuator closer to the base of the cantilever can be seen to have a positive effect on the amount of lateral vibrations as expected. The sample in which the actuator was tucked right up the base of the cantilever so that there was essentially no gap between them further proves that the bridge should be as close to the base of the fiber as possible. This set greatly improved the amount of tip displacement to over 200 µm, achieving the desired level of actuation for the project.

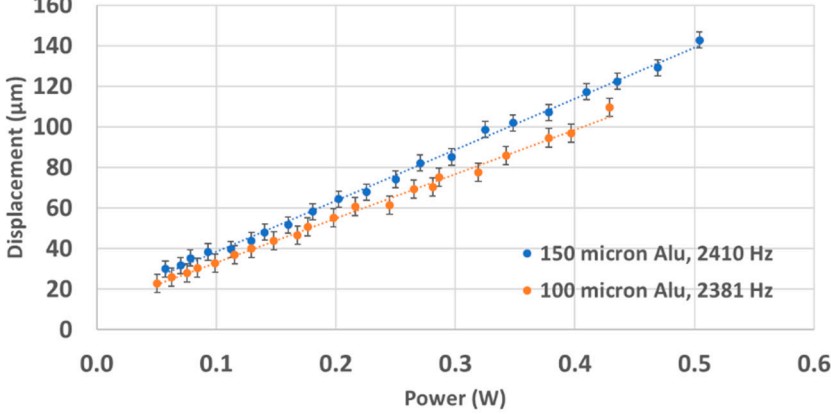

**Figure 10.** Fiber-tip displacement with response to input power for two aluminum samples with different bridge heights (100 and 150 µm).

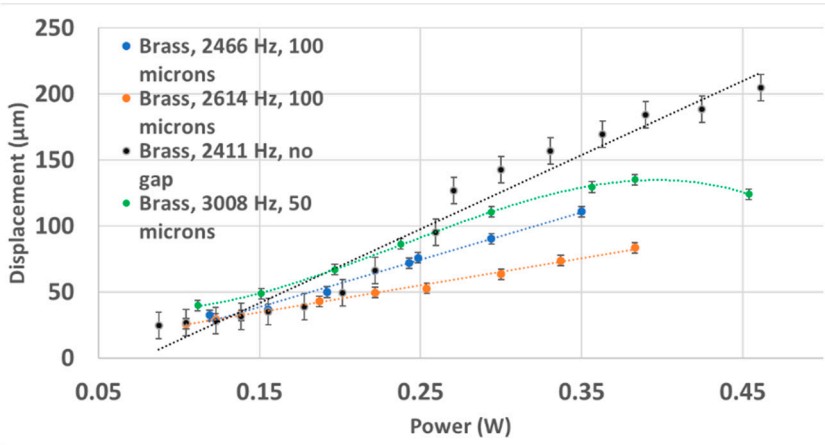

**Figure 11.** Fiber-tip displacement of four 150-μm-tall brass samples emphasizing the effects of natural frequency and distance of the actuator to the cantilever fiber base.

### 4.3. Tip Displacement at Different Duty Cycles

In Section 4.2, it was observed that the actuation bridge was not cooling down as fast as it should during the "off" time cycles, and the average body temperature was drifting upward with higher levels of input power, resulting in reduced overall performance. The cyclic input to all the previous samples was a square signal (i.e., 50% duty cycle). In order to resolve this issue with minimal modifications to the system, the average power was kept the same, but the duty cycle was reduced. This was achieved by reducing the duty cycle but increasing the amplitude of the signal high enough that the area underneath the now narrower signal was the same as the "on" cycle of the old square signal. Note that, while this would result in the same average power, this would still increase the instantaneous power, and this should be balanced and limited to an acceptable level to prevent damaging the thin 20-μm bridge-shaped metal wire. Nevertheless, this reduced duty cycle gives the actuation bridge more time per cycle to cool down. This should counteract some of the negative effects of insufficient cooling. A 150-μm-tall brass-bridge sample with a resonance frequency of 2407 Hz was first run with a 50% duty cycle and then at 20%, and the results are depicted in Figure 12. It can be seen that, by decreasing the duty cycle to 20%, a 53% improvement in performance for the same average power was achieved. This further reinforces the idea that heat dissipation is an issue that needs to be addressed. Nonetheless, this method provides a solution to the problem that would help the system reach the set displacement targets. Lowering the duty cycle resulted in a noticeable net improvement in magnitude of lateral displacement across the board for every single sample tested with it. However, the amount of improvement was not consistent across different samples.

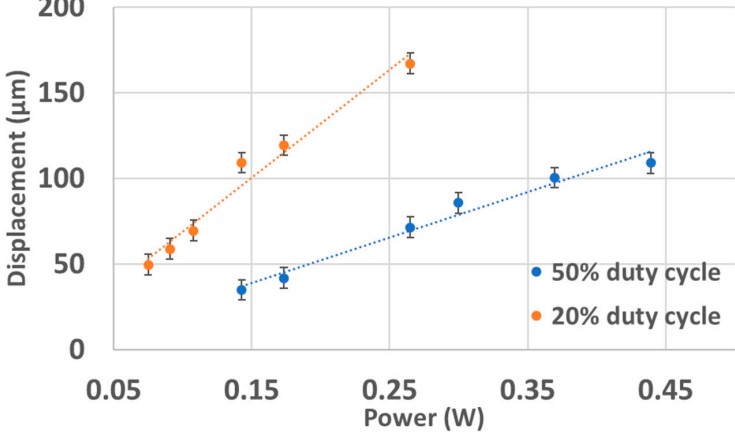

**Figure 12.** Optical-fiber-tip displacement of the same brass-bridge sample under different pulse duty cycles.

### 4.4. Bridge Temperature Measurements

As discussed in the simulation section, it is important to know how hot the actuator system is getting, because it has the potential to compromise the structural integrity and performance of the system. Elevated temperatures can cause the adhesives and SU-8 collars (used to fix everything in place) to soften and, thus, increase damping in a system that relies on resonance. Furthermore, a very high steady-state temperature raises the bottom or floor temperature during oscillations, and the smaller this temperature differential gets, the smaller the actuator oscillations become, which reduces the performance. To measure the temperature of the bridge empirically, two 150-µm-tall brass metal foils were covered in a blob of thermal paste (arctic silver mx-4) with a diameter of about 300 µm. A custom k-type thermocouple tip made of 40-gauge wires was inserted into the ball of carbon-based thermal paste that encased the bridge. The power through the bridge was gradually ramped up, stopping at every interval to make sure the temperatures was at a steady state before recoding the temperature. Note that the temperature measured is not exactly the same as the location of the bridge; it represents the average temperature around the bridge, albeit a bit cooler because of conduction to the environment via the surface of the paste.

Figure 13 shows the temperature measurements collected from the two brass samples along with the simulation results. For the simulation results, the simulation set-up was the same as in Section 3, but the material type was changed to Brass 7030. To simulate the average temperature measured around the bridge, a point on the brass thin foil about 1 mm away was chosen. The minor temperature difference (2–3 °C) between the two samples can be attributed to slightly different thermocouple placement, or thermal paste volume. The measured temperature profile was also consistent with the simulation result. At the upper limits of the system (0.6 W or 1.4 A for the 0.3-Ω brass actuators), the metal foil had an average temperature of approximately 60–70 °C. This means that, over time, at higher levels of power, the adhesive holding the parts together can reach those temperatures as well. This is of concern because temperatures of >60 °C can soften the SU-8 polymer collar structures used to fix the backend. The heat-cured epoxy by Master Bond should, however, remain hard and be structurally sound.

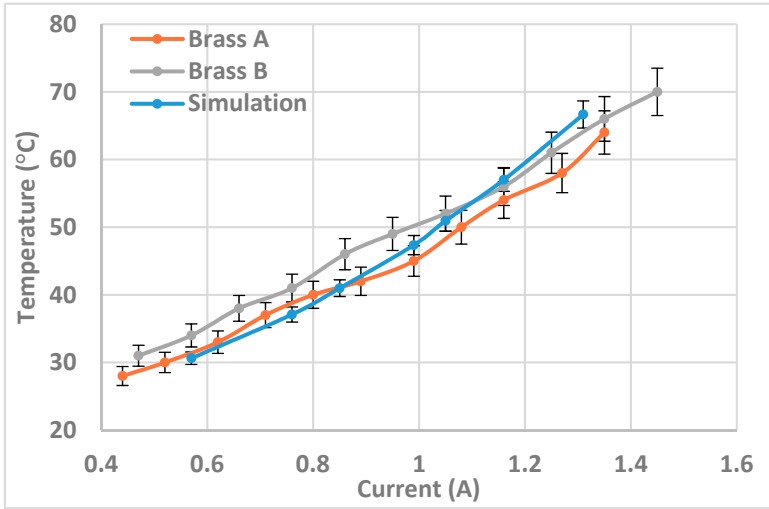

**Figure 13.** Brass foil temperature readings of two 150-µm-tall brass actuators compared to simulation results.

### 4.5. Discussion

This section of the study sets out to assess the performance of the thermally actuated micro-cantilever design and evaluate the magnitude of lateral tip displacement achievable at sufficiently high enough resonance frequency to meet the performance goals set for the project.

The findings show that the design is capable of getting to the target displacement values, and the behavior of the prototypes largely agrees with simulation results and expectations. The tip displacement increased linearly with increased cyclic thermal base excitation, and longer bridges resulted in larger magnitudes of displacement; the bridge acted as a low-quality factor resonator. In addition, decreasing the distance of the bridge and the micro-cantilever resulted in larger amplitudes. The coefficients of determination of all the results presented in this section also reaffirm that the level of performance is a linear function of input power, at least until heat dissipation becomes an issue. In the tests, the performance of different prototypes with the same configuration varied slightly. It is important to note and emphasize that every single prototype was entirely fabricated and assembled by hand and the tolerances were on the order of a few microns. These deviations can arise from many factors. Variations in the angle of the actuating bridge, the height of the fiber relative to the bridge, spread or amount of glue applied, the dimensions of the fiber itself (remember each fiber was also custom-made), distance of the fixed end of fiber to the bridge, size and length of the actuator, bridge materials, how well aligned the fiber is, how rigid the adhesive/platform is, etc. can often have a tangible impact on the performance and behavior of that particular sample. Nevertheless, while the results of samples with the same configuration were not exactly the same, they were generally consistent and predictable. Therefore, it can be concluded that this design has the potential to match the requirements set for this project in terms of tip displacement and resonance frequency, and it should be possible to increase the amount of displacement by optimizing the design parameters and addressing the issues of heat dissipation and structural rigidity. A microscopic image of the tip displacement of the brass sample with no gap between the actuator and the base of the cantilever fiber is shown in Figure 14.

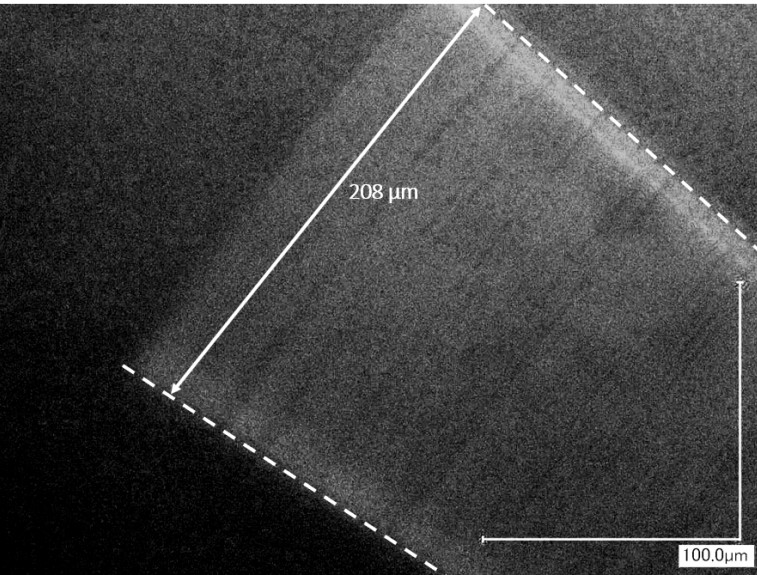

**Figure 14.** Tip displacement of the 2411-Hz brass sample at 0.461 W (with an 8° compensation due to viewing angle).

Lowering the duty cycle while maintaining the same average power was shown to be capable of improving a sample's performance from insufficient to usable at a certain power level. This is especially evident in samples that have a relatively higher resonance frequency. The increased performance is due to fact that, during the "off" states, the bridge has more time to cool down and, thus, it drops down to a lower value than it would have otherwise, which results in a lower average body temperature and a higher bridge temperature differential. Lowering the duty cycle even further should, in theory, improve the results; however, in practice, because of the massive increase in instantaneous power through the actuator wire, it can damage it. Therefore, in samples with lowered duty cycles, the cost of

increased performance has to be balanced against the need for longevity and reliability. The bridge can easily burn or melt when the current running through it becomes too high [27–29]. This is known as the fusing current, and it caps the amount of power that can be given to prototypes.

At high levels of current input (~1 A), most of the devices could run continuously for 10 minutes, and cumulatively for approximately five hours. After about five hours of continuous operation, there was a noticeable drop in performance. This was narrowed down to micro-cracks forming in the EP17 adhesive. The device is expected to operate within the operable temperature range of the adhesive (−50 °C to 340 °C) as recorded from the simulations earlier. The localized hot spots can be generated within the adhesive, which, along with the thermal fatigue due to cyclic temperature change at high frequency, may be the cause of these micro-cracks in adhesive. These micro-cracks, only 1–3 μm wide, compromise the structural integrity of the adhesive and make it so that it can move and play under the induced vibrations. When these cracks form, the back portion of the fiber, i.e., the "fixed" end, is no longer fixed in place, which massively increases the damping coefficient of the system and decreases the magnitude of tip displacement, effectively destroying that particular prototype. This is something that needs to be addressed if the future prototypes are no longer going to be disposable or need to work for more than a few hours. These cracks can be seen in Figure 15.

The actuation system could be optimized via a detailed dynamic thermo-mechanical analysis based on work proposed by Komeili et al. [23]. Such analysis could assist in investigating the actuation system to ensure reliability over an extended period of actuation, while also considering fatigue of the materials.

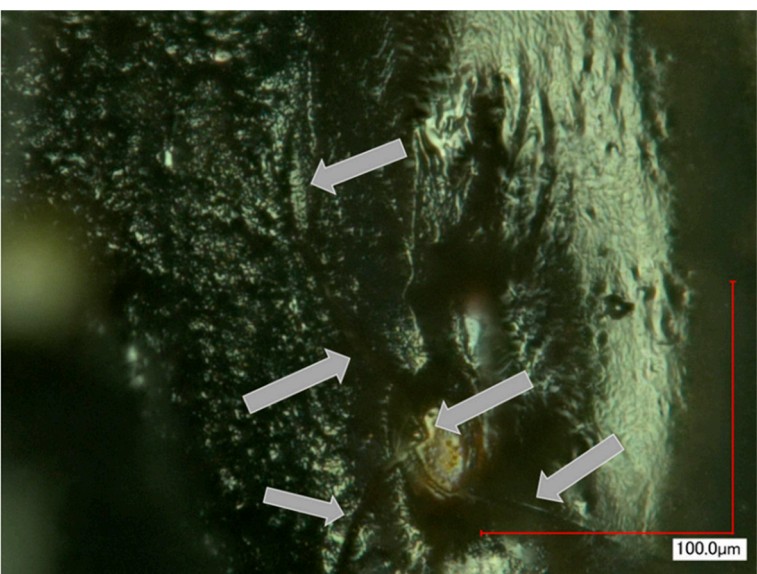

**Figure 15.** The micro-cracks formed in the adhesive after hours of operation.

## 5. Conclusion

Endoscopes are limited by their insertion tube diameter, and many narrow passages inside of the body prevent them from getting to certain organs and areas. The lungs are a notable example of this problem. A novel approach based on exciting a single-mode optical-fiber cantilever at a location close to its base to resonance using a conductive wire was proposed and studied in this paper, in order to design a system that is small enough to be used to image most areas of the lung. The outer diameter of the package and the magnitude of tip displacement and resonance frequency were set as benchmarks for determining the success of this project. The results show that the design can certainly meet the size requirements, producing samples that have an insertion tube diameter of 500 μm (theoretically, even as small as 300 μm) and a distal rigid length of 3–5 mm, well below that which is available at the moment. Table 2 compares some of the properties of the proposed design with the current

available designs. Experimental assessments show that, despite some of the problems outlined, namely insufficient cooling and structural rigidity, the prototypes are still capable of achieving 200 μm of tip displacement at about 2.5 kHz. Although these performance numbers should ideally be higher, they are still sufficient to move the project forward. Finite element analysis was performed to predict the behavior of the electrothermal actuator in terms of magnitude of lateral vibrations and temperature. Experimental results supported the design validity and proved the simulated model. Future iterations of this design will focus on the problem areas identified, including an optimization of shape and size of the thin foil cut-out to better manage and disperse heat, ways to improve the rigidity of the back end of the set-up to reduce damping, and ways to improve repeatability and radiality of the design during manufacturing.

**Table 2.** Device dimensions of the proposed device compared to existing SFE.

|  | **Current SFE** | **Proposed SFE** |
| --- | --- | --- |
| Diameter (mm) | ~1.2 | ~0.5 |
| Length of distal rigid part (mm) | ~10 | 3–5 |

**Author Contributions:** Conceptualization, C.M.; methodology, Y.L. and C.M.; software, M.K.; validation, A.A.A. and M.K.; formal analysis, M.K.; investigation, A.A.A., M.K., Y.L., and P.L.; resources, C.M.; data curation, A.A.A. and M.K.; writing—original draft preparation, A.A.A., M.K., and Y.L.; writing—review and editing, A.A.A., M.K., and C.M.; visualization, M.K.; supervision, C.M. and P.L.; project administration, C.M. and P.L.; funding acquisition, C.M. and P.L.

**Funding:** This work was supported by the Natural Sciences and Engineering Research Council (NSERC) of Canada, Canadian Institutes of Health Research (CIHR), and the Canada Research Chair (CRC) program.

**Conflicts of Interest:** The authors declare no conflict of interest. The funders had no role in the design of the study; in the collection, analyses, or interpretation of data; in the writing of the manuscript, or in the decision to publish the results.

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
