# Peer review of "An Electro-Thermal Actuation Method for Resonance Vibration of a Miniaturized Optical-Fiber Scanner for Future Scanning Fiber Endoscope Design"

_actuators, doi:10.3390/act8010021_

Round 1
Reviewer 1 Report
About units, micrometers and its abbreviation are used interchangeably. It is suggested to use only one notation. In addition, in many cases, there are problems with the abreviations (only meter (m) appears). Check lines: 94, 95, 105, 259, 262-264.
In the graphs, the names of the axes must start with a capital letter.
About the contents:
Why force and Stress are not part of your parameters of interest?
In my experience, when other devices are tested in real conditions, adhesives have serious problems, because they could be in situations where forces and conditions are stronger than they were expected. It could be useful to indicate some of the possible factors provided by real conditios thar could deteriorated to the adhesives.
Finally, please can you added a table in your discussion section, with a comparison of your parameters values with some others currently available, in order to appreciate more clearly the level of differences.
In general, it is a very interesting work, with a high potential impact.
Author Response
Point 1: About units, micrometers and its abbreviation are used interchangeably. It is suggested to use only one notation. In addition, in many cases, there are problems with the abbreviations (only meter (m) appears). Check lines: 94, 95, 105, 259, 262-264.
Response 1: We thank the reviewer for their comment. We followed the reviewer’s suggestion and only used one notation. The abbreviations were also fixed as recommended.
Point 2: In the graphs, the names of the axes must start with a capital letter.
Response 2: Names and axes were corrected as recommended.
Point 3: Why force and Stress are not part of your parameters of interest?
Response 3: This paper focused on the thermal investigations as the authors considered this to be the most innovative and challenging problem of the actuation system. The cantilever undergoes a 0.2mm tip displacement and the stress induced is therefore quite small considering that the cantilever is made from glass (optical fiber). We amended the paper to suggest further analyses related to the dynamic thermo-mechanical behavior of the actuator as we acknowledge that such in-depth investigations could assist optimizing the design of the actuator.
Point 4: In my experience, when other devices are tested in real conditions, adhesives have serious problems, because they could be in situations where forces and conditions are stronger than they were expected. It could be useful to indicate some of the possible factors provided by real conditions that could deteriorated to the adhesives.
Response 4: While we selected an adhesive which has high glass transition temperature (~235°C), 80-90 Shore D hardness, good thermal conductivity (1.4423 W/m.K), and very good electric insulator even at high temperatures, the authors acknowledge that adverse conditions may affect the mechanical/thermal properties of the adhesive. While the adhesive may poorly perform outside these ranges, the authors expect that the device will operate within the operable range of temperature (-50°c to 340°C) as recorded from the simulations. In a practical scenario, some localized hot spots with temperature outside the operable range can be generated, which along with the thermal fatigue may be the cause of micro-cracks in the adhesive (reported in the discussions of the amended article). We believe that the other factors such as humidity, may not constitute serious conditions for potential deterioration of the adhesive given the cantilever to be in a sealed micro-chamber (currently an unsealed needle) in future applications when the technology will be used during endoscopy.
We amended the paper to include a short discussion about the limitations of adhesives in the discussion section.
Point 5: Finally, please can you add a table in your discussion section, with a comparison of your parameters’ values with some others currently available, in order to appreciate more clearly the level of differences.
Response 5: We amended the paper with a short table at the end, comparing our device with the currently available, as recommended. The parameters such as field of view, spatial resolution will be further evaluated in the future design with the lens and optical setup combined with the current design.

Reviewer 2 Report
The paper describes a method and a setup of an optical fiber scanner with electrothermal actuator for resonant excitation to be used in an endoscope. It covers a nice application using an interesting approach including details on design, modelling, fabrication, and test, which will find its interest in the community. However, the presentation of the paper could be improved. In general, the text is not as brief and accurate as possible and many figures are not carefully prepared. Some deficiencies, which have to be removed before publication, are listed below:
1) Thin-film piezoelectric actuators might be an alternative solution for the described cantilever actuation task, which should be mentioned and referred to in the Introduction (U. Sökmen, et al. "Evaluation of resonating Si cantilevers sputter-deposited with AlN piezoelectric thin films for mass sensing applications", J. Micromech. Microeng. 20 (2010) 064007 (7pp); doi:10.1088/0960-1317/20/6/064007).
2) Results are shown for operating the cantilever fiber in air (Figs. 6, 9-12, 14). However, in the human body (organ) probably operation in liquid (body fluid) will be required, leading to much higher damping. How large will the deflection be in that case?
3) Where is the thermal actuator in the schematic in Fig. 1?
4) The photographs (Figs. 3, 4, and 14) have bad contrast and are insufficiently labelled, e.g., where is the collar structure in Fig. 3c)? So, it is very difficult to extract meaningful information from them. E.g., how are the photographs in Figs. 3 b) and c) related to the schematic in a)? Furthermore, in this schematic the important part comprising the metal bridge should be shown enlarged with indications of the used geometric parameters width, thickness, and height. The same applies to Fig. 5, where the part of the bridge is not visible. Furthermore, labelling is too small here.
5) There are several typos, e.g. the digit symbol is used for multiplication instead of the Greek cross, e.g. on page 4 in the Table and the context. Furthermore, on page 2, 3, 8, … for the values of numerous length parameters the micron symbol is missing. “DCF” in line 92, page 2 has to be defined. On page 11, line 351 there is a useless comma. On the same page, line 385 “mircon”. In the reference list, line 585, “catnilever”. Should it read “hypodermic” (page 2, line 82, page 3, line 114, …) or “hypodermal” (page 6, line 229, …)?
6) The abscissa of Fig. 8 has a wrong label: time instead of current.
7) In Figs. 8, and 10-13 typical error bars are lacking and should be added.
Author Response
Point 1: Thin-film piezoelectric actuators might be an alternative solution for the described cantilever actuation task, which should be mentioned and referred to in the Introduction (U. Sökmen, et al. "Evaluation of resonating Si cantilevers sputter-deposited with AlN piezoelectric thin films for mass sensing applications", J. Micromech. Microeng. 20 (2010) 064007 (7pp); doi:10.1088/0960-1317/20/6/064007).
Response 1: We thank the reviewer for their comment. The introduction was amended, and the suggested article was cited.
Point 2: Results are shown for operating the cantilever fiber in air (Figs. 6, 9-12, 14). However, in the human body (organ) probably operation in liquid (body fluid) will be required, leading to much higher damping. How large will the deflection be in that case?
Response 2: We amended the paper describing that none of the components will come into contact with any fluids while inside of the body because the hypodermic needle will be capped (which also holds the lens system at the tip). Only the outer housing will touch any organs or fluids.
Point 3: Where is the thermal actuator in the schematic in Fig. 1?
Response 3: We amended the Fig. 1 as we realized the thermal actuator was not labeled in the original manuscript that was submitted. We believe the amended version is now clear.
Point 4: The photographs (Figs. 3, 4, and 14) have bad contrast and are insufficiently labelled, e.g., where is the collar structure in Fig. 3c)? So, it is very difficult to extract meaningful information from them. E.g., how are the photographs in Figs. 3 b) and c) related to the schematic in a)? Furthermore, in this schematic the important part comprising the metal bridge should be shown enlarged with indications of the used geometric parameters width, thickness, and height. The same applies to Fig. 5, where the part of the bridge is not visible. Furthermore, labelling is too small here.
Response 4: Figs. 3, 4, and 14 were amended as recommended. The only modification we were not able to implement was the contrast of the figure. Specifically, figures 4 b) and c) and figure 15 are composite images produced by taking numerous microscopic shots at ~5-20-micron depth intervals which are then combined together to create 3D images that have ~1-2 mm features. The blurriness and contrast problems are due to shifting light and diffraction at different depths.
Point 5: There are several typos, e.g. the digit symbol is used for multiplication instead of the Greek cross, e.g. on page 4 in the Table and the context. Furthermore, on page 2, 3, 8, … for the values of numerous length parameters the micron symbol is missing. “DCF” in line 92, page 2 has to be defined. On page 11, line 351 there is a useless comma. On the same page, line 385 “mircon”. In the reference list, line 585, “catnilever”. Should it read “hypodermic” (page 2, line 82, page 3, line 114, …) or “hypodermal” (page 6, line 229, …)?
Response 5: We thank the reviewer for pointing out these typos. The paper was amended as recommended.
Point 6: The abscissa of Fig. 8 has a wrong label: time instead of current.
Response 6: The paper was amended as recommended.
Point 7: In Figs. 8, and 10-13 typical error bars are lacking and should be added.
Response 7: The paper was amended as recommended.

Round 2
Reviewer 2 Report
The authors took almost all my suggestions and recommendations into account for an improvement of the paper and explained why photographs could not be improved.